# The Changing Environment in Postgraduate Education in Orthopedic Surgery and Neurosurgery and Its Impact on Technology-Driven Targeted Interventional and Surgical Pain Management: Perspectives from Europe, Latin America, Asia, and The United States

**DOI:** 10.3390/jpm13050852

**Published:** 2023-05-18

**Authors:** Kai-Uwe Lewandrowski, John C. Elfar, Zong-Ming Li, Benedikt W. Burkhardt, Morgan P. Lorio, Peter A. Winkler, Joachim M. Oertel, Albert E. Telfeian, Álvaro Dowling, Roth A. A. Vargas, Ricardo Ramina, Ivo Abraham, Marjan Assefi, Huilin Yang, Xifeng Zhang, Jorge Felipe Ramírez León, Rossano Kepler Alvim Fiorelli, Mauricio G. Pereira, Paulo Sérgio Teixeira de Carvalho, Helton Defino, Jaime Moyano, Kang Taek Lim, Hyeun-Sung Kim, Nicola Montemurro, Anthony Yeung, Pietro Novellino

**Affiliations:** 1Center For Advanced Spine Care of Southern Arizona, 4787 E Camp Lowell Drive, Tucson, AZ 85719, USA; 2Department of Orthopaedics, Fundación Universitaria Sanitas, Bogotá 111321, Colombia; 3Department of Orthopaedic Surgery, College of Medicine—Tucson Campus, Health Sciences Innovation Building (HSIB), University of Arizona, 1501 N. Campbell Avenue, Tower 4, 8th Floor, Suite 8401, Tucson, AZ 85721, USA; openelfar@gmail.com; 4Departments of Orthopaedic Surgery and Biomedical Engineering, College of Medicine—Tucson Campus, Health Sciences Innovation Building (HSIB), University of Arizona, 1501 N. Campbell Avenue, Tower 4, 8th Floor, Suite 8401, Tucson, AZ 85721, USA; lizongming@arizona.edu; 5Wirbelsäulenzentrum/Spine Center—WSC, Hirslanden Klinik Zurich, Witellikerstrasse 40, 8032 Zurich, Switzerland; benedikt.burkhardt@gmail.com; 6Advanced Orthopaedics, 499 E. Central Pkwy, Ste. 130, Altamonte Springs, FL 32701, USA; mloriomd@gmail.com; 7Department of Neurosurgery, Charite Universitaetsmedizin Berlin, 13353 Berlin, Germany; prof.peter.winkler@gmail.com; 8Klinik für Neurochirurgie, Universitätsdes Saarlandes, Kirrberger Straße 100, 66421 Homburg, Germany; joachim.oertel@uks.eu; 9Department of Neurosurgery, Rhode Island Hospital, The Warren Alpert Medical School of Brown University, Providence, RI 02903, USA; atelfeian@lifespan.org; 10Orthopaedic Surgery, University of São Paulo, Brazilian Spine Society (SBC), Ribeirão Preto 14071-550, Brazil; adowling@dws.cl (Á.D.); hladefin@fmrp.usp.br (H.D.); 11Department of Neurosurgery, Foundation Hospital Centro Médico Campinas, Campinas 13083-210, Brazil; rothvargas@hotmail.com; 12Neurological Institute of Curitiba, Curitiba 80230-030, Brazil; ramina@hospitalinc.com.br; 13Clinical Translational Sciences, University of Arizona, Roy P. Drachman Hall, Rm. B306H, Tucson, AZ 85721, USA; abraham@pharmacy.arizona.edu; 14Department of Biology, Nano-Biology, University of North Carolina, Greensboro, NC 27413, USA; massefi@aggies.ncat.edu; 15Orthopaedic Department, The First Affiliated Hospital of Soochow University, No. 899 Pinghai Road, Suzhou 215031, China; suzhouspine@163.com; 16Department of Orthopaedics, First Medical Center, PLA General Hospital, Beijing 100853, China; xifengzhang301@163.com; 17Minimally Invasive Spine Center Bogotá D.C. Colombia, Reina Sofía Clinic Bogotá D.C. Colombia, Department of Orthopaedics Fundación Universitaria Sanitas, Bogotá 0819, Colombia; jframirezl@yahoo.com; 18Department of General and Specialized Surgery, Gaffrée e Guinle University Hospital, Federal University of the State of Rio de Janeiro (UNIRIO), Rio de Janeiro 20270-004, Brazil; fiorellirossano@hotmail.com; 19Faculty of Medecine, University of Brasilia, Federal District, Brasilia 70919-900, Brazil; mauriciogpereira@gmail.com; 20Pain and Spine Minimally Invasive Surgery Service at Gaffre e Guinle University Hospital, Rio de Janeiro 20270-004, Brazil; paulo.carvalho@unirio.br; 21La Sociedad Iberolatinoamericana De Columna (SILACO), and the Spine Committee of the Ecuadorian Society of Orthopaedics and Traumatology (Comité de Columna de la Sociedad Ecuatoriana de Ortopedia y Traumatología), Quito 170521, Ecuador; jaime.moyano7@icloud.com; 22Good Doctor Teun Teun Spine Hospital, Anyang 14041, Republic of Korea; limkat@hanmail.net; 23Department of Neurosurgery, Nanoori Hospital, Seoul 06048, Republic of Korea; neurospinekim@gmail.com; 24Department of Neurosurgery, Azienda Ospedaliero Universitaria Pisana, University of Pisa, 56124 Pisa, Italy; nicola.montemurro@unipi.it; 25Desert Institute for Spine Care, Phoenix, AZ 85020, USA; ayeung@sciatica.com; 26Guinle and State Institute of Diabetes and Endocrinology, Rio de Janeiro 20270-004, Brazil; pietro.novellino@hotmail.com

**Keywords:** postgraduate residence training, orthopedic surgery, neurosurgery, technology advances, simulation, augmented reality, navigation, robotics, artificial intelligence, skill-based training

## Abstract

Personalized care models are dominating modern medicine. These models are rooted in teaching future physicians the skill set to keep up with innovation. In orthopedic surgery and neurosurgery, education is increasingly influenced by augmented reality, simulation, navigation, robotics, and in some cases, artificial intelligence. The postpandemic learning environment has also changed, emphasizing online learning and skill- and competency-based teaching models incorporating clinical and bench-top research. Attempts to improve work–life balance and minimize physician burnout have led to work-hour restrictions in postgraduate training programs. These restrictions have made it particularly challenging for orthopedic and neurosurgery residents to acquire the knowledge and skill set to meet the requirements for certification. The fast-paced flow of information and the rapid implementation of innovation require higher efficiencies in the modern postgraduate training environment. However, what is taught typically lags several years behind. Examples include minimally invasive tissue-sparing techniques through tubular small-bladed retractor systems, robotic and navigation, endoscopic, patient-specific implants made possible by advances in imaging technology and 3D printing, and regenerative strategies. Currently, the traditional roles of mentee and mentor are being redefined. The future orthopedic surgeons and neurosurgeons involved in personalized surgical pain management will need to be versed in several disciplines ranging from bioengineering, basic research, computer, social and health sciences, clinical study, trial design, public health policy development, and economic accountability. Solutions to the fast-paced innovation cycle in orthopedic surgery and neurosurgery include adaptive learning skills to seize opportunities for innovation with execution and implementation by facilitating translational research and clinical program development across traditional boundaries between clinical and nonclinical specialties. Preparing the future generation of surgeons to have the aptitude to keep up with the rapid technological advances is challenging for postgraduate residency programs and accreditation agencies. However, implementing clinical protocol change when the entrepreneur–investigator surgeon substantiates it with high-grade clinical evidence is at the heart of personalized surgical pain management.

## 1. Introduction

Postgraduate medical education is evolving rapidly, driven by technological advancements, changes in healthcare delivery models, and shifting societal and cultural trends [1]. Traditional classroom-based learning is replaced with competency-based education [2,3,4,5,6,7,8], which focuses on developing specific skills and abilities to prepare surgical residents for today’s more complex healthcare environment. Postgraduate medical education is increasingly taking place online [9,10,11]. This trend also applies to practicing surgeons who have long since graduated. Now, online courses and training programs [12,13] sponsored by specialty organizations [14,15] and industry vendors [16] can be accessed from anywhere in the world, making education more accessible and convenient. The COVID-19 pandemic further solidified the use of online programs in postgraduate education [17,18,19,20]. Moreover, there is a growing emphasis on interprofessional collaboration to provide more comprehensive patient care. This collaborative patient-centered approach to healthcare is fueled by medical technology advancements, such as telemedicine, virtual reality [19,21,22,23,24], artificial intelligence applications [25,26,27] with wearable devices for skill-based simulations [28,29,30], electronic health records [31,32,33], and translational knowledge integration—all of which increase the demands on the next generation of orthopedic surgeons and neurosurgeons who now must be versed in the clinical application of these technologies to be effective in the delivery of the best possible care to patients in an increasingly cost-constrained environment characterized by an imbalance in clinical innovation and resource commitment. The authors of this editorialized perspective article came together to highlight the ongoing challenges in their educational programs and how the underlying changes in orthopedic and neurosurgery postgraduate training may impact personalized interventional and surgical pain management specialty care delivery in the future.

## 2. Shifting Trends

The international team of authors of this perspective article—many of whom are directors of postgraduate education programs in orthopedic surgery and neurosurgery—identified the shift toward competency-based training as the most significant trend altering the curriculum and culture of their programs [2,3,4,5,6,7,8]. Nowadays, proof of proficiency in specific skills is required before a trainee can progress to the next level of training [2]. This trend is playing out globally regardless of cultural differences and geographical boundaries. A quick straw poll among the authors also revealed that critical thinking with the application of a broad knowledge base rooted in evidence-based practice is also commonplace. There are, however, a few differences that are worth pointing out.

In Europe, trainees have more opportunities to undertake research projects and present at conferences with time allotted for these activities in specific rotations [18,19,23,29,30,34,35,36]. However, residency programs tend to be longer when compared to the United States, where such provisions are mandated but, in reality, uncommon. In 2020 a multinational survey by the European Association of Neurological Surgeons (EANS) reported a significant decline in surgical exposure during training from the 1970s to 2019 [37,38]. However, the reported results were doubted, and a whitewash of the data was anticipated by others [39]. It has to be assumed that this trend will be further accelerated by the new wage agreement that was passed in Germany in 2020 [38]. This wage agreement now limits the number of on-call services to a maximum of four per month, not only for residents but also for attendings [40]. The European authors of this editorial suggested that introducing new technologies, such as simulation training and virtual reality, to enhance surgical education is necessary and becoming more and more feasible because of enhanced collaboration across departmental and institutional barriers to providing more diverse and comprehensive training opportunities for trainees [29].

In Latin America, participating authors reported that program development is underway to focus on improving access to postgraduate education for surgeons in under-resourced areas [41,42,43]. This expansion is often performed on a national and international basis with the development of regional networks to promote collaboration and knowledge sharing between regional institutions and with institutions in other countries. Several of the coauthors of this perspective worked on improving postgraduate education in Orthopedics and Neurosurgery in that way. Several authors are corresponding foreign members of their country’s respective National Medical Academies. Online resources and Zoom teleconferencing with national and international faculty are heavily employed to overcome geographical barriers and enhance training opportunities [44,45]. These activities paved the way for more structured and standardized training programs in Latin America.

Our Asian coauthors reported rapidly expanding postgraduate training programs to meet the growing demand for surgical services. In southeast Asia and China, in particular, the population growth is forecast to continue until 2050 (Figure 1). The disease burden from painful musculoskeletal and spinal conditions is substantial and will likely rise (Figure 2 and Figure 3). Therefore, greater emphasis on subspecialization in spinal surgery or joint replacement is reported employing simulation training and 3D printing application in new surgical implants. Interinstitutional and international collaboration to gain access to resources and expertise, particularly in under-resourced areas, are attempted but still limited due to the postpandemic geopolitical restrictions on travel and internet access.

In the United States, a similar shift toward competency-based training, with milestones and objective assessments to track trainee progress, was introduced just as in Europe, with an increasing emphasis on patient safety and quality improvement and a greater focus on teamwork and communication skills [2,4,5,6,8].

Surgical education is increasingly technology-dominated by virtual and augmented reality simulators. Newer training pathways have emerged outside the traditional employment models in academia and industry. This trend is also occurring in European countries. For example, nearly 20% of German and Swiss medical school graduates do not start a postgraduate specialty training program but venture into the private sector for job opportunities, suggesting a particular frustration with low job prospects in comparison to the time investment needed to become a licensed physician [46,47]. The changing culture of postgraduate education in orthopedics and neurosurgery is characterized by a greater focus on competency-based training, increased use of technology, and greater collaboration between institutions and countries to provide more diverse and comprehensive training opportunities.

## 3. The Residents’ Perspective

Orthopedic and neurosurgery residents have been polled regarding their experiences with and perceptions of the modern learning environment [48]. Most respondents viewed the simulation and virtual reality technology as positive and opined that it improved their education [49]. However, a subset was concerned about losing autonomy and thought it could diminish the hands-on experience [50]. Previous reports suggest that many residents strongly desire better communication with the attending teaching surgeon with more detailed feedback about their performance, emphasizing the need for a more individualized learning experience. This notion differs from the current technology implementation trend in the postgraduate training process [51,52,53,54]. Being effectively trained and prepared for the real-world working scenarios of orthopedic surgeons and neurosurgeons required that adequate training took precedence over the desire for reduced work hours. However, work–life balance issues are relevant, making a case for more efficient learning scenarios. Hence, residents place value on attending physician mentorship, particularly when applied in hands-on patient encounters. One-on-one teaching experiences are still at the top of the list [55,56].

## 4. The Mentors Perspective

Mentoring is an integral part of any orthopedic and neurosurgery residency program [57]. Academic orthopedic surgeons and key opinion leaders play a crucial role in shaping the professional development of the residents they teach [58]. Conversely, mentors can enhance their knowledge and skills through mentoring, which often gives them insights into new techniques, procedures, and technologies. It may even force them to stay on top of the latest research and changes in up-to-date clinical practice protocols, thus improving the mentor’s professional development. Ideally, this interchange expands the mentor surgeon’s professional network by connecting with other mentors in the residency program and beyond, directly and indirectly, through coaching their resident mentees, potentially striking up new collaborations and opportunities for both the mentor and mentee [51]. As a result, teaching surgeons may find mentoring a rewarding experience with a sense of accomplishment and fulfillment in knowing that they played a role in their resident mentees’ success. Career advancement opportunities may open up for the mentor surgeon as they build a reputation as a respected and knowledgeable leader. As a result, mentor surgeons may be invited for speaking engagements, leadership positions, and other opportunities that can enhance their careers [57].

## 5. Work-Hour Restrictions

Work-hour restrictions have had a significant impact on surgeon training. Work hours for resident doctors in many countries are now mandated. One of the primary goals of work-hour restrictions was to improve resident well-being and reduce burnout and fatigue and improve quality of life [56,59]. Gone are the days were surgeons in training would learn their craft in a few years with higher surgical case volumes in routine practice or during the after-hours call. Today, residents have reduced exposure to patient care and surgical cases. Independent problem-solving and technical skills are harder to acquire because of fewer opportunities to perform surgeries. Hence, senior residents and attending staff physicians have to take on a greater supervisory role in the operating room, which can benefit some trainees but also limits their autonomy and independence. The reduction of resident work hours has also changed the clinical learning environment with the need for greater efficiencies in the learning process [60,61,62]. Where mentorship relationships between attending and resident trainee surgeons used to dominate hands-on skill acquisition, postgraduate programs are now forced to look increasingly at replacing these personal and interactive teaching scenarios [63] with more sophisticated didactic sessions and virtual surgical simulation models [5,30]. While there is no doubt that these new digital educational experiences can be helpful and should be implemented, more information is needed on whether these means of training the next generation of orthopedic surgeons and neurosurgeons are as effective as traditional methods [28]. Many program director authors of this perspective article are worried that virtual reality cannot fully replicate the clinical setting and that some trainees may feel they are missing out on a valuable hands-on experience due to work-hour restrictions.

High-quality postgraduate surgical training, at least in part, is influenced by the opportunity to perform surgeries and acquire experience and skills under the guidance of another proficient surgeon. Limiting patient encounters and surgical training opportunities by law may prompt the need to lower graduation and certification standards or increase the length of the residency to continue to provide the expected high-quality care to patients. To comply with the new wage agreement in Germany, for example, the number of residents eventually has to be increased. On 5 April 2023, a 16% resident work hour reduction was demanded by their union to be contractually mandated for the hospitals in the Kanton Zurich in Switzerland, where surgical residents are expected to work around 40 h a week [64]. Over 20 years ago, Reulen and März delineated that an annual surgery volume of 2100 neurosurgical operations allows training appropriately 7–8 residents [65]. However, the number of surgical procedures in residency programs performed per resident cannot be proportionally increased simultaneously. Unless the respective departments are expanded, residents and attendings will have fewer patient care encounters and less exposure to surgical procedures. Extending the duration of residency programs is openly discussed and has been implemented in some programs to ensure that residents are adequately prepared for independent practice [66]. However, longer training times may deter medical school graduates from becoming orthopedic surgeons or neurosurgeons. However, at this point, it is unclear whether or not this dynamic will impact the certification [67] and credentialing process [68]. Additional unintended consequences may arise from the German 2020 wage law. For example, reducing the earning potential for those surgeons who are used to more than four on-call services per month may drive them outside academic residency programs to look for more lucrative employment opportunities. Additional unintended consequences are possible.

As a result, the surgeon shortages experienced today may become more pronounced as a significant portion of currently practicing surgeons in these subspecialties are over the age of 55 and are thinking about retiring. Patient care could also be negatively impacted. Trainee surgeons may have less time to devote to each patient, which could hollow out the entire postgraduate training experience. There is some evidence to suggest that work-hour restrictions may lead to more errors and complications [61,68,69]. Hence, patient safety may suffer from reduced hands-on experience and longer training duration. The debate on these controversial issues will likely continue and further research is needed to identify the most effective postgraduate training models for the future.

## 6. Examples of Slow Adoption in Postgraduate Training

Delayed innovation implementation in orthopedics and neurosurgery has led to the slower adoption and integration of new technologies, procedures, or practices. For example, minimally invasive surgery (MIS) techniques employing various versions of small tubular and bladed retractor systems have been developed and proven effective in many orthopedic and neurosurgical procedures. Their widespread adoption and implementation in postgraduate training programs were initially slow. MIS procedures gained popularity in the early 2000s. Adoption delays were related to the learning curve of new surgical approaches, the need for specialized training, and the initial capital equipment costs required. It also took a few years until higher-grade clinical evidence emerged, proving their safety and effectiveness compared to traditional surgical techniques. Over time and with more experience, training programs have adapted to include MIS techniques because of the overwhelming benefits of reduced scarring, faster recovery, and improved patient outcomes that are being realized. Endoscopic spine surgery is a similar MIS example where adoption and integration into orthopedic and neurosurgical postgraduate training programs have been even slower. Both the retractor-based and endoscopic techniques emerged around the same time in the 1990s. However, endoscopic spinal decompression surgery is much harder to learn as it requires a higher skill level. The MIS retractor-based surgeries are minimized tissue-sparing versions of traditional translaminar open surgery and, therefore, easier understood by traditionally trained spine surgeons. Endoscopic decompression, mainly through the transforaminal approach, requires becoming accustomed to new surgical access unfamiliar to most spine surgeons and mastering eye–hand coordination integrating hand maneuvers with direct visualization of the surgical site on the video screen. Advances in imaging technology and 3D printing have allowed for the creation of patient-specific implants in orthopedic surgery. These implants are custom-designed to match the patient’s anatomy, resulting in better fit and improved outcomes. However, the implementation of patient-specific implants has been relatively slow due to the need for specialized software, equipment, and additional regulatory considerations. Therefore, it has not permeated postgraduate training programs. Robotics-assisted surgery is another such technology application, which has the potential to enhance precision and accuracy in orthopedic surgery and neurosurgery. However, the high costs associated with acquiring and maintaining robotic systems, as well as the need for specialized training, have contributed to the delayed implementation in many institutions. The added time required to learn the technology is difficult to carve out of an already packed postgraduate residency training program schedule. Regenerative strategies have been in preclinical and clinical trials for over two decades. Autologous chondrocyte harvesting followed by in vitro expansion and reimplantation has been practiced since the 1990s. However, these tissue engineering and cellular therapies did not find widespread use because of the many challenges related to regulatory approval and protocol standardization. Moreover, the limited clinical evidence is weaker than that of total joint replacement. As the benefits and effectiveness of these innovations become more established and the barriers to implementation are addressed, their integration into orthopedic surgery practices is expected to increase, ultimately improving patient outcomes and advancing the field.

## 7. Resident Clinical Research and Solutions

Program directors of this article worry about whether clinical research is still a priority of orthopedic and neurosurgery residents. In the United States, the Accreditation Council for Graduate Medical Education (ACGME) requires that orthopedic and neurosurgery residency programs provide structured educational experiences in research, including training in research methodology and data analysis. The ACGME also mandates that residents complete at least one research project during training and that each resident be provided with at least 60 days of protected time for research. While the pandemic has affected clinical research activities, recent studies found that residents and researchers have adapted to the situation by using virtual platforms to conduct meetings, research activities, and data analysis [20,70]. One particular study found that 83% of resident research thesis projects were published on average approximately 7 years from the start of their residency training. The graduate adjusted H-index was associated with increased success and decreased time to publication, while a lower journal impact factor was associated with taking significantly less time to reach publication. Coming out of the COVID-19 pandemic, clinical research activities are expected to extend from the virtual to in-person interactive platforms on the benchtop and clinical levels.

Research rotations may also provide residents with the skill set to quickly understand and adopt new technologies. Preparing the future generation of surgeons to have the aptitude to keep up with the rapid technological advances may be challenging for postgraduate residency programs but critical to solving the lag problem between innovation and implementation. Another solution to the fast-paced innovation cycle in orthopedic surgery and neurosurgery includes adaptive learning skills to seize opportunities for innovation with execution and implementation by facilitating translational research and clinical program development across traditional boundaries between clinical and nonclinical specialties.

## 8. Impact of Transformative Technologies and Targeted Care Models

The emerging technologies likely to impact orthopedic and neurosurgery residency programs within the next five years include robotics, augmented reality, 3D printing, artificial intelligence, nanotechnology, and regenerative technologies with stem cells and their respective stimulatory and growth factors. Robotic surgery has improved accuracy in total joint replacement component placement [71,72,73]. Navigation is another add-on technology with similar goals of improving patient outcomes by reducing inaccuracies in the surgical approaches and techniques thought to be the source of higher complication and revision surgery rates [74,75,76,77,78]. Depending on the application, navigation and augmented reality (AR) may be utilized separately or together [79,80,81,82]. Both technologies aim at more accurately navigating complex structures during surgery. Artificial intelligence (AI) may improve diagnostic accuracy and treatment plans based on large amounts of patient data to determine which painful or tumorous conditions may benefit from intervention and which not—a break with traditional laboratory or imaged-based medical necessity criteria for surgery [26,83,84,85]. AI could also assist surgeons during surgery, providing real-time feedback and guidance [86,87,88]. Three-dimensional printing will further facilitate and simplify manufacturing-customized implants and prosthetics [89,90,91]. The academic research conducted within postgraduate residency programs could show whether it cost-effectively improves clinical outcomes and lowers complication and revision rates [90]. Alternatively, 3D printing may also be used to create models of a patient’s anatomy to assist training programs in planning, teaching, and simulating complex surgeries. Nanotechnology may be applied in the targeted delivery of drugs and other therapeutic agents to treat affected tissues directly [92,93]. Regenerative technologies with the utilization of stem cells [94] and their respective growth factors have the potential to play a significant role in the future of orthopedic surgery and neurosurgery with nerve, cartilage, and bone regeneration to help with bone defects, spinal cord injuries, and peripheral nerve damage [95]. Tissue engineering technologies in conjunction with 3D printing [95] could play into regenerative strategies by producing newly formed replacement tissues. Stem cells may also play a role in chronic pain management [95,96,97]. These trends highlight the need for postgraduate training programs to incorporate education on these new technologies that provide the basis for more personalized and targeted care models aimed at the structural correlate causing the patient’s pain. Interventional and surgical pain management that incorporates these emerging technologies will likely improve clinical outcomes while reducing costs via lower complication and reoperation rates. On the other hand, the question is how these transformative technologies and targeted care models can be adopted and deployed in a cost-responsible and affordable, yet financially enabling, way.

## 9. Discussion

Emerging technologies will probably change the surgical indications for many painful chronic degenerative conditions and thus impact the array of surgical techniques taught in academic orthopedic and neurosurgery postgraduate residency programs as traditional methods are increasingly replaced by modern less burdensome and more targeted procedures. They may form the basis for future sound stewardship principles in public healthcare systems. This expected significant impact on the training of the next generation of orthopedic surgeons and neurosurgeons will require specialized knowledge of these advanced technologies and new skills as these technologies become more widely used in clinical practice. Thus, they will become part of the standard postgraduate residency training curriculum. Moreover, many of these technologies require collaboration between orthopedics, engineering, biology, neurosurgery, and many other disciplines. Hence, postgraduate residency programs, both on the clinical and research sides, need to adapt to this more multidisciplinary training approach by facilitating collaborations with healthcare professionals and researchers from different fields. The growing emphasis on regenerative medicine and other advanced technologies also requires the next-generation orthopedic surgeon and neurosurgeon to be more versed in basic and clinical research. Benchtop research skills and basic concepts of clinical trial design and execution, research methodology, data analysis, and other skills necessary for conducting clinical trials and other research studies, including skills in applying for and securing research funding, will have to be taught at a more sophisticated level. Finally, postgraduate training programs in orthopedic surgery and neurosurgery must nurture a culture of innovation consistent with common societal values of community advancement. The onus of research into new ways to treat painful conditions must fall on more than just scientists or those with specialized training outside the clinical arena. Instead, the key features of innovation must be ingrained in our training programs. One place to start is quality control and monitoring, where regulations are much more forgiving for the institution of new ideas in the care protocols surrounding patients. However, this type of innovation cannot be the sole purview of clinician educators. If left to quality measures alone, we would have efficient protocols but no new treatments. The fast-moving and constantly evolving nature of orthopedics and neurosurgery and the emergence of advanced technologies are likely to accelerate the innovation cycle. For this reason, the next generation of orthopedic surgeons and neurosurgeons will require the skills to stay up to date with the latest developments in their fields.

Residents have a demanding and often rigorous training schedule that can leave little time for personal pursuits. Balancing clinical duties, research, family obligations, and test preparation can be challenging and require excellent time-management and prioritization skills. The amount of time orthopedic residents have to accomplish these tasks can vary depending on various factors, including the specific residency program, the number of hours worked per week, and the resident’s work pace and efficiency. However, most residency programs struggle with allotting enough time for research and academic pursuits outside clinical duties. Still, the amount of time available may be limited. Program directors need to be prepared to teach some of these skills and, in some cases, act not just as mentors but as life coaches to whom resident trainees look up for advice and guidance on navigating the complex and rigorous orthopedic and neurosurgery training curriculum.

The future orthopedic surgeon and neurosurgeon involved in personalized surgical pain management will need to be versed in several disciplines, such as bioengineering, computers, basic research, clinical research including trial designs, epidemiological research, public health policy development, and economic accountability. Adaptive learning skills are needed to seize opportunities for innovation with execution and implementation by facilitating translational research and clinical program development across traditional boundaries between clinical and nonclinical specialties. Preparing the future generation of surgeons to have the aptitude to keep up with rapid technological advances is challenging for postgraduate residency programs and accreditation agencies. However, implementing clinical protocol change when the entrepreneur–investigator surgeon substantiates it with high-grade clinical evidence is at the heart of personalized surgical pain management.

## 10. Conclusions

The objectives in the orthopedic and neurosurgical residency core curriculum programs are changing mainly in response to the changes in healthcare delivery models and the rapid emergence of new technologies. Teaching programs are shifting emphasis to more skill- and competency-based teaching methods. Work-hour restrictions and attention to work–life balance issues may improve the mental health of residents by lowering burnout rates but necessitate higher efficiencies in teaching methodologies. Advanced AR simulation and 3D modeling techniques may be helpful, [98] but one-on-one interaction with their mentors in a postgraduate residency training program remains crucial. Residents should work closely with their program directors and mentors to establish a schedule that allows for efficient use of their time while still meeting the demands of their clinical and academic responsibilities. [99] They should also prioritize their tasks and delegate obligations where possible to ensure they can complete their duties without sacrificing their personal or professional goals. Ultimately, the ability to balance these various demands will vary from individual to individual and depend on multiple factors, including their work ethic, time-management skills, and support systems. Residency programs have to teach the next generation of orthopedic surgeons and neurosurgeons the life-long learning skills to position themselves effectively in the future personalized care models of surgical pain management. 

## Figures and Tables

**Figure 1 jpm-13-00852-f001:**
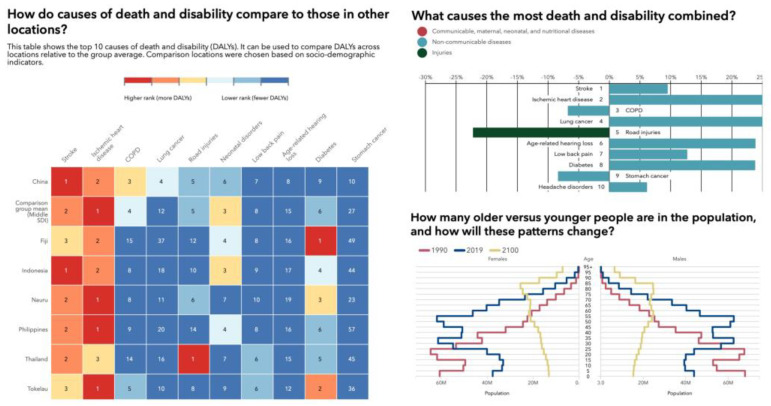
Illustration of disease burden expressed in death and disability (DALYs) based on sociodemographic indicators across several locations in Southeast Asia and China relative to the group average, showing an aging population and a change in the order of the top 10 causes of death and disability (DALYs) in 2019 and percent change from 2009 to 2019 for all ages combined. In 2019, musculoskeletal conditions and low back pain were of much higher relevance to public healthcare systems than in 2009. *Source*: Institute for Health Metrics Evaluation. Used with permission. All rights reserved.

**Figure 2 jpm-13-00852-f002:**
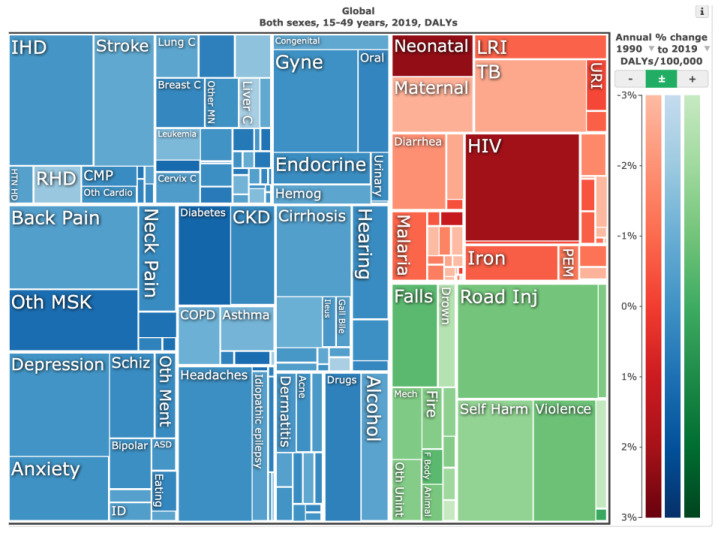
Illustrative tree map of causes and global disease burden expressed in years lived with disability (YLDs) for both genders ages 15 to 49 years in 2019. YLDs for low back pain was 8.19% (6.3–10.19%), other musculoskeletal diseases 7.57% (5.54–10.08%), osteoarthritis 4.12% (2.41–7.65%), neck pain 2.12% (1.36–3.22%), falls 2.59% (2.28–2.97%), road injury 1.51% (1.37–1.67%), exposure to mechanical forces 0.88% (0.69–1.14%), interpersonal violence 0.49% (0.42%–0.56%), and other unintentional injuries 0.37% (0.3–0.47%). *Source*: Institute for Health Metrics Evaluation. Used with permission. All rights reserved.

**Figure 3 jpm-13-00852-f003:**
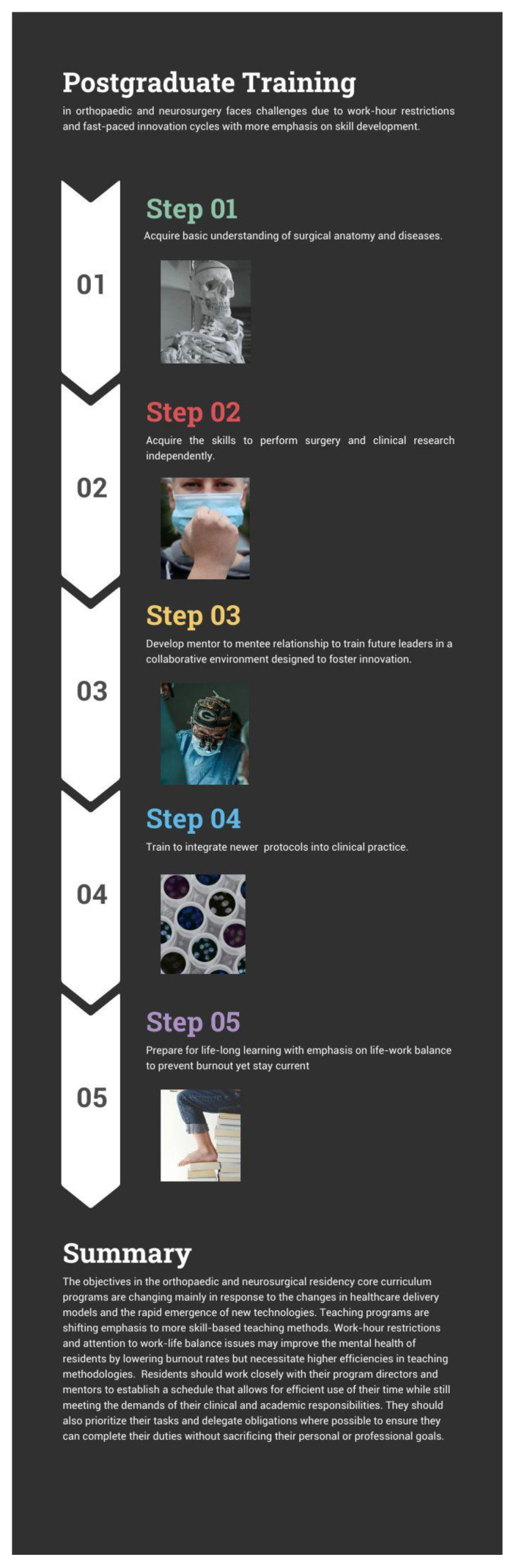
Infographic on shifting emphasis in orthopedic and neurosurgery residency program.

## Data Availability

The data presented in this study are publicly available.

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
