# Peer review of "The Changing Environment in Postgraduate Education in Orthopedic Surgery and Neurosurgery and Its Impact on Technology-Driven Targeted Interventional and Surgical Pain Management: Perspectives from Europe, Latin America, Asia, and The United States"

_jpm, 2023, doi:10.3390/jpm13050852_

Round 1
Reviewer 1 Report
To improve the manuscript, the author could provide more specific examples of the challenges and opportunities discussed. For instance, the author could provide examples of the specific technological advancements that are influencing education in orthopaedic and neurosurgery. Additionally, the author could provide specific suggestions for addressing the challenges identified in the abstract. For instance, the author could suggest ways to improve the efficiency of postgraduate residency programs to keep pace with the fast-paced flow of information and the rapid implementation of innovation. The author could also provide examples of interdisciplinary collaborations that have successfully facilitated translational research and clinical program development across traditional boundaries between clinical and non-clinical specialties.
Author Response
Reviewer #1
To improve the manuscript, the author could provide more specific examples of the challenges and opportunities discussed. For instance, the author could provide examples of the specific technological advancements that are influencing education in orthopaedic and neurosurgery. Additionally, the author could provide specific suggestions for addressing the challenges identified in the abstract. For instance, the author could suggest ways to improve the efficiency of postgraduate residency programs to keep pace with the fast-paced flow of information and the rapid implementation of innovation. The author could also provide examples of interdisciplinary collaborations that have successfully facilitated translational research and clinical program development across traditional boundaries between clinical and non-clinical specialties.
Response:
We appreciate this reviewer’s response as it pointed out weakness in our perspective article that needed addressing. We added examples of the innovation-implementation lag problem and potential solutions to the abstract. We also expanded the manuscript by adding a half-page section (section 6) entitled “Examples of Slow Adoption in Postgraduate Training”. In addition, section 7 headline was amended to read “7. Resident Clinical Research & Solutions”. We hope that the additional language added to the manuscript satisfied this reviewer’s concern so that the manuscript can be accepted for publication. We would be glad to expand the manuscript if deemed appropriate by the reviewers and editors.
The text additions are as follows:
“6. Examples of Slow Adoption in Postgraduate Training
Delayed innovation implementation in orthopedic and neurosurgery led to the slower adoption and integration of new technologies, procedures, or practices. For ex-ample, minimally invasive surgery (MIS) techniques employing various versions of small tubular and bladed retractor systems have been developed and proven effective in many orthopedic and neurosurgical procedures. Their widespread adoption and implementa-tion in postgraduate training programs were initially slow. MIS procedures gained popularity in the early 2000s. Adoption delays were related to the learning curve of new surgical approaches, the need for specialized training, and the initial capital equipment costs required. It also took a few years until higher-grade clinical evidence emerged, proving its safety and effectiveness compared to traditional surgical techniques. Over time and with more experience and training programs adapted to include MIS techniques because of the overwhelming, the benefits of reduced scarring, faster recovery, and improved patient outcomes are being realized. Endoscopic spine surgery is a similar MIS example where adoption and integration into orthopedic and neurosurgical postgraduate training programs have been even slower. Both the retractor-based and endoscopic techniques emerged around the same time in the 1990ies. However, endoscopic spinal decompression surgery is much harder to learn as it requires a higher skill level. The MIS retractor-based surgeries are minimized tissue-sparing versions of traditional trans-laminar open surgery and, therefore, easier understood by traditionally trained spine surgeons. Endoscopic decompression, mainly through the transforaminal approach, requires getting accustomed to new surgical access unfamiliar to most spine surgeons and mastering eye-hand coordination integrating hand maneuvers with direct visualization of the surgical site on the video screen. Advances in imaging technology and 3D printing have allowed for the creation of patient-specific implants in orthopaedic surgery. These implants are custom-designed to match the patient's anatomy, resulting in better fit and improved outcomes. However, the implementation of patient-specific implants has been relatively slow due to the need for specialized software, equipment, and additional regulatory considerations. Therefore, it has not permeated postgraduate training pro-gramms. Robotics-Assisted Surgery is another such technology application which has the potential to enhance precision and accuracy in orthopaedic and neurosurgery surgery. However, the high costs associated with acquiring and maintaining robotic systems, as well as the need for specialized training, have contributed to the delayed implementation in many institutions. The added time required to learn the technology is difficult to carve out of an already packed postgraduate residency training program schedule. Regener-ative strateges have been in preclinical and clinical trials for over two decades. Autologous chondrocyte harvesting followed by in vitro expansion and reimiplantation has been practiced since the 1990ies. However, these tissue engineering and cellular therapies did not find widespread use because of many regulatory approval, and protocol standardi-zation challenges. Moreover, the limited clinical evidence is weaker than that in total joint replacement. As the benefits and effectiveness of these innovations become more estab-lished and the barriers to implementation are addressed, their integration into ortho-paedic surgery practices is expected to increase, ultimately improving patient outcomes and advancing the field.
- Resident Clinical Research & Solutions
Program directors of this article worry about whether clinical research is still a priority of orthopaedic and neurosurgery residents. In the United States, the Accreditation Council for Graduate Medical Education (ACGME) requires that orthopaedic and neu-rosurgery residency programs provide structured educational experiences in research, including training in research methodology and data analysis. The ACGME also man-dates that residents complete at least one research project during training and that each resident must be provided with at least 60 days of protected time for research. While the pandemic has affected clinical research activities, recent studies found that residents and researchers have adapted to the situation by using virtual platforms to conduct meetings, research activities, and data analysis [20, 71]. One particular study found that 83% of resident research thesis projects were published on average approximately 7 years from the start of their residency training. Graduate adjusted H-index was associated with in-creased success and decreased time to publication, while a lower journal impact factor was associated with taking significantly shorter time to reach publication. Coming out of the COVID-19 pandemic, clinical research activities are expected to extend from the virtual to in-person interactive platforms on the benchtop and clinical levels.
Research rotations may also provide residents with the skill set to quickly under-stand and adopt new technologies. Preparing the future generation of surgeons to have the aptitude to keep up with the rapid technological advances may be challenging for postgraduate residency programs but is critical to solving the lag problem between in-novation and implementation. Another solution to the fast-paced innovation cycle in orthopedic- and neurosurgery includes adaptive learning skills to seize opportunities for innovation with execution and implementation by facilitating translational research and clinical program development across traditional boundaries between clinical and non-clinical specialties.”
Reviewer 2 Report
Thank you for the opportunity to review this manuscript.
Here, the authors provide an editorialized perspective article relevant to individuals in postgraduate residency programs, specifically in orthopedic surgery and neurosurgery. It addresses the evolving nature of medical education, the impact of technology, and the need for interdisciplinary skills in the modern healthcare landscape.
The authors draw on their expertise and experience to identify key trends and issues and offer recommendations for improving postgraduate education. The manuscript is well-written, with clear and concise language, and is supported by relevant references. Additionally, while the authors offer recommendations for improving postgraduate education, it is unclear how feasible or practical these recommendations are in practice.
Strengths:
- Relevance: The abstract addresses a timely and relevant topic - the impact of personalized care models and technological advancements on the education and training of orthopedics and neurosurgeons.
- Comprehensive scope: The manuscript covers various aspects, including augmented reality, simulation, navigation, robotics, and artificial intelligence, highlighting the need for interdisciplinary skills.
- Recognition of challenges: It acknowledges the difficulties postgraduate residency programs face in keeping up with rapid technological advances and work-life balance issues.
- Emphasis on evidence-based practice: It stresses the importance of implementing clinical protocol changes supported by high-grade clinical evidence in personalized surgical pain management.
Weaknesses:
- Lack of specific examples: The manuscript discusses various technologies and innovations but does not provide specific examples or evidence of their impact on surgical education.
- Limited focus: The manuscript primarily focuses on the challenges and needs of orthopedic and neurosurgery residents, potentially overlooking the broader context of other surgical specialties.
- Lack of practical recommendations: While it highlights the importance of adaptive learning and interdisciplinary skills, the abstract does not offer concrete suggestions for how these can be effectively integrated into postgraduate residency programs.
Author Response
Reviewer #2
“Thank you for the opportunity to review this manuscript.
Here, the authors provide an editorialized perspective article relevant to individuals in postgraduate residency programs, specifically in orthopedic surgery and neurosurgery. It addresses the evolving nature of medical education, the impact of technology, and the need for interdisciplinary skills in the modern healthcare landscape.
The authors draw on their expertise and experience to identify key trends and issues and offer recommendations for improving postgraduate education. The manuscript is well-written, with clear and concise language, and is supported by relevant references. Additionally, while the authors offer recommendations for improving postgraduate education, it is unclear how feasible or practical these recommendations are in practice.
Strengths:
Relevance: The abstract addresses a timely and relevant topic - the impact of personalized care models and technological advancements on the education and training of orthopedics and neurosurgeons.
Comprehensive scope: The manuscript covers various aspects, including augmented reality, simulation, navigation, robotics, and artificial intelligence, highlighting the need for interdisciplinary skills.
Recognition of challenges: It acknowledges the difficulties postgraduate residency programs face in keeping up with rapid technological advances and work-life balance issues.
Emphasis on evidence-based practice: It stresses the importance of implementing clinical protocol changes supported by high-grade clinical evidence in personalized surgical pain management.
Weaknesses:
Lack of specific examples: The manuscript discusses various technologies and innovations but does not provide specific examples or evidence of their impact on surgical education.
Limited focus: The manuscript primarily focuses on the challenges and needs of orthopedic and neurosurgery residents, potentially overlooking the broader context of other surgical specialties.
Lack of practical recommendations: While it highlights the importance of adaptive learning and interdisciplinary skills, the abstract does not offer concrete suggestions for how these can be effectively integrated into postgraduate residency programs.”
Response:
This reviewers’ comments are similar to the first reviewers. Therefore, we will respond in a similar way. Once more, we appreciate this reviewer’s response as it pointed out weakness in our perspective article that needed addressing. We added examples of the innovation-implementation lag problem and potential solutions to the abstract. We also expanded the manuscript by adding a half-page section (section 6) entitled “Examples of Slow Adoption in Postgraduate Training”. In addition, section 7 headline was amended to read “7. Resident Clinical Research & Solutions”. We hope that the additional language added to the manuscript satisfied this reviewer’s concern so that the manuscript can be accepted for publication. We would be glad to expand the manuscript if deemed appropriate by the reviewers and editors.
“6. Examples of Slow Adoption in Postgraduate Training
Delayed innovation implementation in orthopedic and neurosurgery led to the slower adoption and integration of new technologies, procedures, or practices. For ex-ample, minimally invasive surgery (MIS) techniques employing various versions of small tubular and bladed retractor systems have been developed and proven effective in many orthopedic and neurosurgical procedures. Their widespread adoption and implementa-tion in postgraduate training programs were initially slow. MIS procedures gained popularity in the early 2000s. Adoption delays were related to the learning curve of new surgical approaches, the need for specialized training, and the initial capital equipment costs required. It also took a few years until higher-grade clinical evidence emerged, proving its safety and effectiveness compared to traditional surgical techniques. Over time and with more experience and training programs adapted to include MIS techniques because of the overwhelming, the benefits of reduced scarring, faster recovery, and improved patient outcomes are being realized. Endoscopic spine surgery is a similar MIS example where adoption and integration into orthopedic and neurosurgical postgraduate training programs have been even slower. Both the retractor-based and endoscopic techniques emerged around the same time in the 1990ies. However, endoscopic spinal decompression surgery is much harder to learn as it requires a higher skill level. The MIS retractor-based surgeries are minimized tissue-sparing versions of traditional trans-laminar open surgery and, therefore, easier understood by traditionally trained spine surgeons. Endoscopic decompression, mainly through the transforaminal approach, requires getting accustomed to new surgical access unfamiliar to most spine surgeons and mastering eye-hand coordination integrating hand maneuvers with direct visualization of the surgical site on the video screen. Advances in imaging technology and 3D printing have allowed for the creation of patient-specific implants in orthopaedic surgery. These implants are custom-designed to match the patient's anatomy, resulting in better fit and improved outcomes. However, the implementation of patient-specific implants has been relatively slow due to the need for specialized software, equipment, and additional regulatory considerations. Therefore, it has not permeated postgraduate training pro-gramms. Robotics-Assisted Surgery is another such technology application which has the potential to enhance precision and accuracy in orthopaedic and neurosurgery surgery. However, the high costs associated with acquiring and maintaining robotic systems, as well as the need for specialized training, have contributed to the delayed implementation in many institutions. The added time required to learn the technology is difficult to carve out of an already packed postgraduate residency training program schedule. Regener-ative strateges have been in preclinical and clinical trials for over two decades. Autologous chondrocyte harvesting followed by in vitro expansion and reimiplantation has been practiced since the 1990ies. However, these tissue engineering and cellular therapies did not find widespread use because of many regulatory approval, and protocol standardi-zation challenges. Moreover, the limited clinical evidence is weaker than that in total joint replacement. As the benefits and effectiveness of these innovations become more estab-lished and the barriers to implementation are addressed, their integration into ortho-paedic surgery practices is expected to increase, ultimately improving patient outcomes and advancing the field.
- Resident Clinical Research & Solutions
Program directors of this article worry about whether clinical research is still a priority of orthopaedic and neurosurgery residents. In the United States, the Accreditation Council for Graduate Medical Education (ACGME) requires that orthopaedic and neu-rosurgery residency programs provide structured educational experiences in research, including training in research methodology and data analysis. The ACGME also man-dates that residents complete at least one research project during training and that each resident must be provided with at least 60 days of protected time for research. While the pandemic has affected clinical research activities, recent studies found that residents and researchers have adapted to the situation by using virtual platforms to conduct meetings, research activities, and data analysis [20, 71]. One particular study found that 83% of resident research thesis projects were published on average approximately 7 years from the start of their residency training. Graduate adjusted H-index was associated with in-creased success and decreased time to publication, while a lower journal impact factor was associated with taking significantly shorter time to reach publication. Coming out of the COVID-19 pandemic, clinical research activities are expected to extend from the virtual to in-person interactive platforms on the benchtop and clinical levels.
Research rotations may also provide residents with the skill set to quickly under-stand and adopt new technologies. Preparing the future generation of surgeons to have the aptitude to keep up with the rapid technological advances may be challenging for postgraduate residency programs but is critical to solving the lag problem between in-novation and implementation. Another solution to the fast-paced innovation cycle in orthopedic- and neurosurgery includes adaptive learning skills to seize opportunities for innovation with execution and implementation by facilitating translational research and clinical program development across traditional boundaries between clinical and non-clinical specialties.”